# Metformin intervention ameliorates AS in *ApoE-/-* mice through restoring gut dysbiosis and anti-inflammation

Ning Yan[1,2☯], Lijuan Wang[1,3☯], Yiwei Li[4☯], Ting Wang[4], Libo Yang[1,2], Ru Yan[2,5], Hao Wang[4]*, Shaobin Jia[2,5]*

**1** Clinical Medical College, Ningxia Medical University, Yinchuan, China, **2** Heart Centre & Department of Cardiovascular Diseases, General Hospital of Ningxia Medical University, Yinchuan, China, **3** Department of Cardiovascular Diseases, The Second Hospital of Yinchuan, Yinchuan, Ningxia, China, **4** Department of Pathogenic Biology and Medical Immunology, School of Basic Medical Sciences, Ningxia Medical University, Yinchuan, Ningxia, China, **5** Ningxia Key Laboratory of Vascular Injury and Repair Research, Ningxia Medical University, Yinchuan, China

☯ These authors contributed equally to this work.
* jsbxn@163.com (SJ); wanghaograduate@126.com (HW)

**Data Availability Statement:** All 16s rRNA sequencing raw files are available from the Sequences Read Archive database at the NCBI (accession number PRJNA624814).

## Abstract

Atherosclerosis (AS) is closely associated with chronic low-grade inflammation and gut dysbiosis. Metformin (MET) presents pleiotropic benefits in the control of chronic metabolic diseases, but the impacts of MET intervention on gut microbiota and inflammation in AS remain largely unclear. In this study, *ApoE-/-* mice with a high-fat diet (HFD) were adopted to assess the MET treatment. After 12 weeks of MET intervention (100mg·kg$^{-1}$·d$^{-1}$), relevant indications were investigated. As indicated by the pathological measurements, the atherosclerotic lesion was alleviated with MET intervention. Moreover, parameters in AS including body weights (BWs), low-density lipoprotein (LDL), triglyceride (TG), total cholesterol (TC) and malondialdehyde (MDA) were elevated; whereas high-density lipoprotein (HDL) and total superoxide dismutase (T-SOD) levels were decreased, which could be reversed by MET intervention. Elevated pro-inflammatory interleukin (IL)-1β, IL-6, tumor necrosis factor (TNF)-α and lipopolysaccharide (LPS) in AS were decreased after MET administration. However, anti-inflammatory IL-10 showed no significant difference between AS group and AS +MET group. Consistently, accumulated macrophages in the aorta of AS were conversely lowered with MET treatment. The results of 16S rRNA sequencing and analysis displayed that the overall community of gut microbiota in AS was notably changed with MET treatment mainly through decreasing *Firmicutes*, *Proteobacteria*, *Romboutsia*, *Firmicutes/Bacteroidetes*, as well as increasing *Akkermansia*, *Bacteroidetes*, *Bifidobacterium*. Additionally, we found that microbiota-derived short-chain fatty acids (SCFAs) including acetic acid, propionic acid, butyric acid and valeric acid in AS were decreased, which were significantly upregulated with MET intervention. Consistent with the attenuation of MET on gut dysbiosis, decreased intestinal tight junction protein zonula occludens-1 (ZO)-1 in AS was restored after MET supplementation. Correlation analysis showed close relationships among gut bacteria, microbial metabolites SCFAs and inflammation. Collectively, MET intervention ameliorates AS in *ApoE-/-* mice through restoring gut dysbiosis and anti-inflammation, thus

**Funding:** This work was supported by the Key Research and Development Projects of Ningxia, China (Grant number 2018BEG02006); the Natural Science Foundation of Ningxia, China (Grant number 2021AAC05010); the National Natural Science Foundation, China (Grant number 82060057) and the Natural Science Foundation of Ningxia, China (Grant number 2018AAC03257). The funders had no role in study design, data collection and analysis, decision to publish, or preparation of the manuscript.

**Competing interests:** The authors have declared that no competing interests exist.

can potentially serve as an inexpensive and effective intervention for the control of the atherosclerotic cardiovascular disease.

## Introduction

Atherosclerosis (AS) maintains a leading cause of death worldwide despite considerable advances in prevention, diagnosis and therapy [1, 2]. Patients with AS were characterized by angina, peripheral arterial disease, lipid metabolism disorders, inflammatory response and endothelial dysfunction [3, 4]. Dyslipidemia, hypertension, diabetes, an unhealthy lifestyle and genetic factors have been considered as the primary drivers in AS, but the exact mechanism remains poorly understood [5], thus novel effective interventions against AS are urgently needed. Growing evidences have demonstrated that gut dysbiosis is closely linked to the progression of AS [6, 7]. A study indicates that the relative abundance of *Collinsella* genus in the intestinal of AS patients are increased, while *Roseburia* and *Eubacterium* are decreased compared to healthy individuals [8]. Regulation of the composition of the overall gut microbiota by increasing *Bacteroidetes* and *Akkermensia* abundance, as well as reducing *Firmicutes* and *Proteobacteria* abundance can prevent AS in *ApoE$^{-/-}$* mice [9]. Thus, the modulation of the gut microbiome may contribute to improving the disease.

Numerous studies have demonstrated that persistent low-grade inflammation plays a critical role in the development and complications of AS, with elevated interleukin (IL)-6 and tumor necrosis factor (TNF)-α [10–12]. Moreover, lipid metabolic disorders were considered to the major cause of AS [13]. Experimental and clinical studies have reported the close relationships between hypercholesterolemia and AS [14, 15]. Excessive low-density lipoprotein (LDL) in the intima of arteries induces the infiltration and activation of inflammatory cells to release inflammatory indicators, leading to the impairment of endothelial functions in the progression of AS [16]. Dysbacteriosis of AS exacerbates gut barrier injury to increase the permeability and reduce the integrity, ultimately cause the lipopolysaccharide (LPS) translocation from the mucosa into the vascular circulation for triggering an inflammatory cascade reaction by activating macrophages (Mψs) via LPS-Toll-like receptor (TLR)-4 pathway [17].

In addition to LPS, gut microbial short-chain fatty acids (SCFAs) are thought to be closely involved in the regulation of insulin resistance, lipid metabolism, and inflammatory status in chronic metabolic diseases [18]. Emerging studies have demonstrated that SCFAs exhibit a wide range of functions from immune regulation to metabolism in a variety of tissues and organs [19, 20]. SCFAs have differential effects on the activation of endothelial Nod-like receptor protein 3 (NLRP3) inflammasome and related carotid AS progression [21]. SCFAs mainly including propionate, acetate, and butyrate in the cecum were significantly decreased in 13-week high-fat cholesterol-fed *Ldlr$^{-/-}$*(*Casp1$^{-/-}$*) compared with *Ldlr$^{-/-}$* mice [22]. SCFAs have been regarded as potential indicators in AS progression. The understanding of the underlying mechanisms related to inflammation in the pathogenesis of AS raises the opportunities in the control of this disease.

Metformin (MET), a biguanide agent, has been widely proven to show pleiotropic effectiveness in diabetes and AS including hypoglycemic activity, ameliorating endothelial dysfunction, and lipid metabolic disorders [23]. However, the effects of MET on gut microbial community and inflammation in AS remain largely undetermined. In this study, we aimed to investigate the effect of MET intervention on gut microbiota and inflammation in *ApoE$^{-/-}$* mice. Our study was aiming to contribute to the further understanding of the role of MET on the complicated interactions among gut microbiota, inflammation and metabolism in AS progression.

## Materials and methods

### Animals and diets

Eight-week male $ApoE^{-/-}$ mice (20–22 g) were purchased from Vital River Laboratory Animal Technology Co., Ltd., Beijing, China (Product Number: scxk2016-0006). All mice were bred and housed at 22±2°C under 12 h light/12 h dark cycle with free access to water and food at the Experiment Animal Centre of Ningxia Medical University. The mice were housed in cages with up to 5 animals and acclimated to their environment before the experiment. A high-fat diet (HFD) with 1.25% cholesterol (60% fat, 20% carbohydrate, 20% protein, No.TP28520) was purchased from TROPHIC Animal Feed High-tech Co., Ltd., Nantong, China. All animal experiments were approved by the Ethics Committee of the General Hospital of Ningxia Medical University (No.2016-106).

### Experimental design

As displayed in the diagram of this study (**Fig 1A**), after three weeks of an adaption, $ApoE^{-/-}$ mice were randomly assigned to three groups (10 mice/group): (a) control group (CON) with a normal diet; (b) atherosclerosis group (AS) with a HFD diet [24]; (c) AS with MET group (AS+MET) were administered with oral 100mg/kg/day MET (Roche Pharmaceuticals, USA) as previous descriptions [25, 26]. Body weights or food intake were respectively monitored weekly or every 2 days during the experiment. After 12 weeks of intervention, feces samples were freshly acquired for the subsequent 16S rRNA sequencing. All mice were euthanized by 4% sodium pentobarbital and sacrificed for the further study.

### Assessment of AS lesion

Quantification of the atherosclerotic lesion was performed by calculating the lipid deposition size in the aortic sinus using oil red O staining as previously described [27]. Briefly, hearts together with a short segment of the aorta were harvested and embedded in optimal cutting temperature compound. Quick-frozen on 4–6 μm cryostat sections were taken from the left ventricular outflow tract where the 3 aortic valves first appeared up to where the aortic valves disappeared and were collected on glass slides and then stained with oil red O. The oil red O stained area of the atherosclerotic lesion was observed using the microscope (Olympus, Japan). The percentage of lesion area of the aortic sinus was analyzed by Image J software (National Institutes of Health, USA). The average value (mean of three sections per mouse) was measured by a single observer blinded to the experimental protocol and used for quantitative evaluation.

Masson's trichrome staining was used to measure the vascular lesion. Briefly, the frozen sections were conventionally dewaxed into the water and stained with the prepared Weigert iron hematoxylin for 5–10 min. The sections were differentiated with acidic ethanol differentiation solution and washed with water, and then returned to blue with Masson's blue solution and wash with water. After washing with distilled water for 1 min, the sections were dyed with ponceau red magenta staining solution for 5–10 min. In the above operation process, a weak acid working solution was prepared according to the ratio of distilled water: weak acid solution = 2:1, and the sections were washed with a weak acid working solution for 1 min. After washing the phosphomolybdic acid solution for 1–2 min, the sections were washed with the prepared weak acid working solution for an additional 1 min. Then the sections were directly put into the aniline blue staining solution for 1 to 2 min. After washing with the prepared weak acid working solution for 1 min, 95% ethanol was quickly used to dehydrate the sections. The sections were dehydrated with anhydrous ethanol for 3 times (5–10 sec/time).

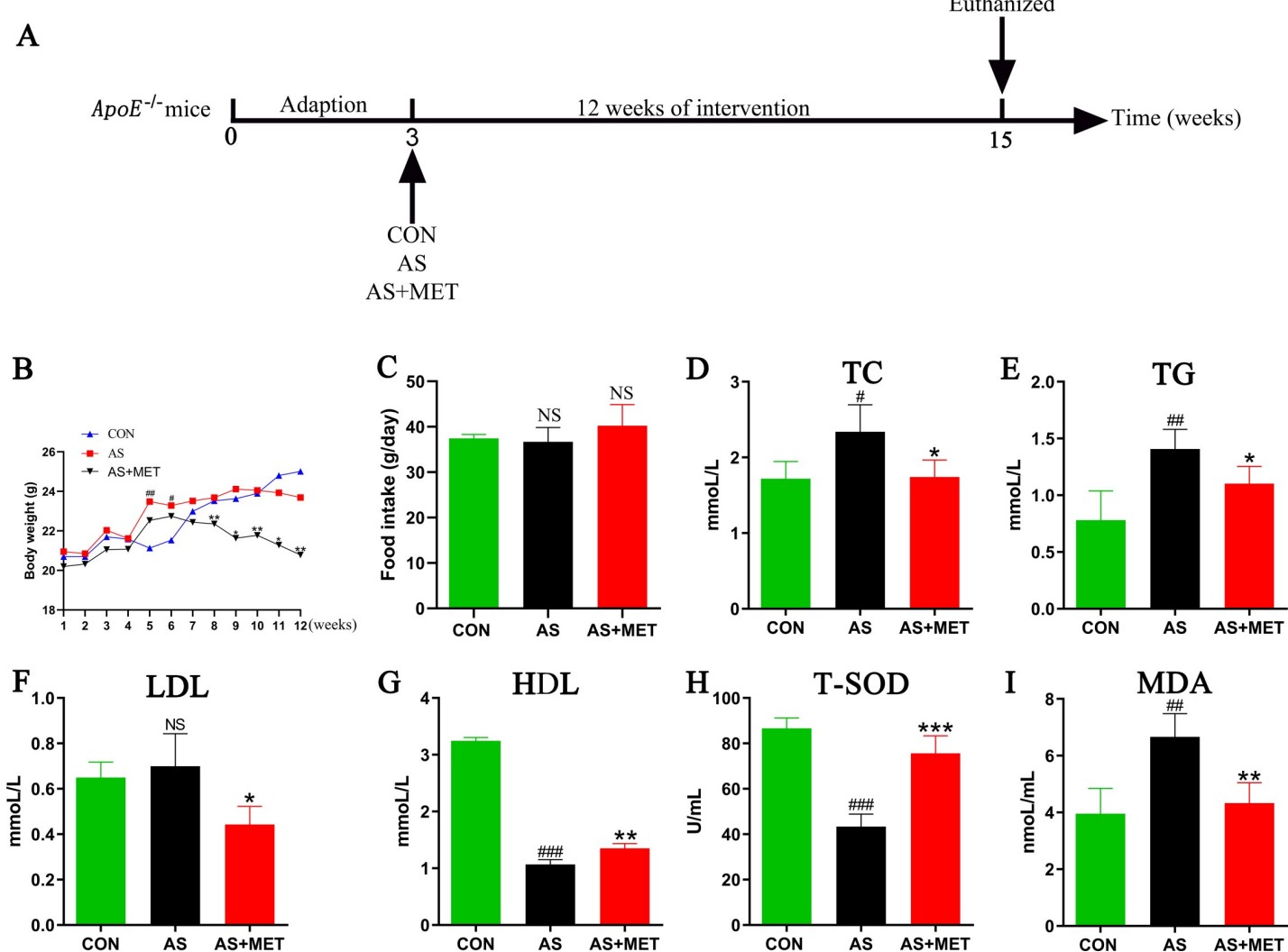

**Fig 1. Effects of oral MET intervention on the parameters in AS. (A)** Schematic diagram of the study. After 3 weeks of adaption feeding, *ApoE⁻/⁻* mice with 18-20g were randomly assigned into 3 groups: control group (CON), atherosclerosis group (AS) and AS treated with metformin (MET) group (AS+MET). After 12 weeks of intervention, all of the mice were sacrificed and related indicators were investigated; **(B)** Body weights (BWs) dynamic curve; **(C)** BWs at the end of the experiment; **(D)** Total cholesterol (TC); **(E)** Triglyceride (TG); **(F)** High-density lipoprotein (HDL); **(G)** Low-density lipoprotein (LDL); **(H)** Total superoxide dismutase(T-SOD); **(I)** Malondialdehyde (MDA). Results were from 4 independent experiments were performed in triplicate. Data were presented as mean ± SEM. #(CON vs. AS): #$p < 0.05$, ##$p < 0.01$, ###$p < 0.001$. *(AS vs. AS+MET): *$p < 0.05$, **$p < 0.01$, ***$p < 0.001$. NS, no significant difference.

Transparent sections with xylene for 3 times (1–2 min/time) were covered with a coverslip and neutral gum seal and then observed under the Olympus microscope.

Hematoxylin and eosin (HE) staining was performed to measure pathological changes. After mice euthanasia, aorta roots were isolated, fixed in formalin, dehydrated, and then embedded in paraffin. HE staining continuous slides were observed by the blinders of this experiment using the microscope (Olympus, Japan).

**Fecal DNA extraction, 16S ribosomal RNA (16S rRNA) sequencing.** Alterations of gut microbiota were determined by high throughput sequencing and analysis of fecal microbial 16S rRNA [28]. At the end of the experiment, 5 mice from each group were randomly selected to obtain fresh feces samples. The DNA was extracted from 200 mg samples using the cetyltrimethy-lammonium bromide (CTAB) method [29]. The DNA concentration and purity were identified by 1.0% agarose electrophoresis. After the adjustment of DNA concentration to 1 ng/μL, the

16SrRNA sequencing was performed [28–30]. Briefly, 16S rRNA genes were amplified by using V3-V4 regions bacterial primers (341F 5'- CCTAYGGGRBGCASCAG-3' and 806R 5'- GG ACTACNNGGGTATCTAAT-3'). All PCR reactions were carried out with Phusion® High-Fidelity PCR Master Mix (New England Biolabs, USA). Sequencing libraries were generated using the Ion Plus Fragment Library Kit 48 rxns (Thermo Scientific, USA). The library quality was assessed on the Qubit@ 2.0 Fluorometer (Thermo Scientific, USA).The library was sequenced on an Illumina HiSeq 2500 platform (Illumina, USA) by Beijing Nuo He Zhi Yuan Technology Co., Ltd., China. All raw sequences have been submitted to the Sequences Read Archive (SRA) database at the NCBI with an accession number PRJNA624814.

## Determination of plasma endotoxin

LPS levels in plasma from diverse groups were determined using the Limulus amebocyte lysate kit (Xiamen Bioendo Technology Co.Ltd, Xiamen, China) according to the manufacturer's instruction.

## Measurements of plasma lipid profiles, oxidative stress and inflammation levels

Plasma levels of triglycerides (TG), total cholesterol (TC), high-density lipoprotein (HDL) and low-density lipoprotein (LDL) were measured by an automatic biochemical analyzer (AU400 Olympus, Japan).

Plasma activity of total superoxide dismutase (T-SOD) and malondialdehyde (MDA) were respectively detected by commercial kits (Nanjing Jiancheng Bioengineering Ins., Nanjing, China) [29].

Inflammatory IL-1β, IL-6, TNF-α and IL-10 levels in plasma and aorta root were determined using BD™ cytometric bead array (CBA) mouse inflammatory cytokine kits by flow cytometer (Accuri™ C6 BD, USA), and then the concentrations were calculated by FCAP Array software (BD Bioscience, USA) [28].

## Immunofluorescence staining

To determine changes of inflammatory cells in aorta and gut barrier permeability, aorta macrophages (Mψs) and tight junctional zonula occludens (ZO)-1 in diverse groups were measured by immunofluorescence staining. In brief, sections were deparaffinized, then slides were incubated with methanol/water (1:1) containing 0.3% hydrogen peroxide to quench the endogenous peroxidase activity. After 10% goat serum for 30 min at room temperature to remove the nonspecific binding, sections were respectively probed with rat anti-mouse F4/80 (1:250 dilution, Abcam, ab6640, USA) or anti-ZO-1 antibody (1:200 dilution, Santa Cruz Biotechnology, sc-33725, USA) overnight at 4˚C. Then samples were incubated with secondary antibody fluorescein (FITC)-conjugated goat anti-rat IgG (H+L) (1:500 dilution, Proteintech, SA00003-11, USA) for 1 h at room temperature. Sections on coverslips were mounted with a sealer containing DAPI (ZSGB-BIO, ZLI-9557, China). Images were captured in a blinded manner with a Leica DMI3000+ DFC310FX fluorescence microscope (Leica, Germany). The positive areas in plaque were quantified by Image-Pro Plus 6.0.

## Determination of fecal short-chain fatty acids (SCFAs)

As crucial end-products of gut microbiota, fecal SCFAs were determined using the gas chromatography-mass spectrometer (GC-MS) method with an Agilent's MSD ChemStation (Agilent, USA) as previously described [31].

## Statistical analysis

GraphPad Prism 6 (GraphPad, USA) and SPSS 23.0 (IBM, UK) was used for statistical analyses. Quantitative data were displayed as the mean ± SEM (standard error of the mean), which were analyzed by one-way ANOVA followed by Tukey's multiple comparisons test. Difference between two groups was assessed by student's $t$ test while the data meeting Gaussian distribution, if not, nonparametric tests were used. Spearman's correlation analyses was used to evaluate the associations among gut microbiota, SCFAs, and inflammatory indicators. All experiments were performed in triplicate. P values with less than 0.05 were considered statistically significant.

## Results

### Routine parameters in diverse groups

There was no significant difference in initial BWs among three groups. However, we found that BWs in AS group was significantly elevated compared to the CON group in week 5 and 6, whereas BWs of AS mice were obvious decreased after MET intervention from week 8 to week 12, demonstrating that MET intervention could attenuate weight gain in AS (p<0.01; **Fig 1B**). There's no significant change in food intake among diverse groups (p>0.05; **Fig 1C**).

To evaluate the effects of MET administration on the lipid metabolism in atherosclerotic $ApoE^{-/-}$ mice, TC, TG, LDL and HDL levels in plasma were respectively examined. Compared to the CON group, plasma TC (p<0.05; **Fig 1D**) and TG (p<0.01; **Fig 1E**) in AS group were notably increased, but LDL (p>0.05; **Fig 1F**) showed no alteration, whereas HDL (p<0.001; **Fig 1G**) level was dramatically reduced. After MET administration, TC, TG and LDL levels in AS were significantly down-regulated (**Fig 1D–1F**), as well as HDL (p<0.01; **Fig 1G**) was elevated, suggesting that long-term supplementation of MET may ameliorate atherosclerotic lipid disorders.

To identify the effects of MET intake on oxidative stress levels in $ApoE^{-/-}$ mice, we detected the levels of serum T-SOD (p<0.001; **Fig 1H**) and MDA (p<0.01; **Fig 1I**). T-SOD level in AS model was elevated in comparison with the CON group, which was rectified by MET treatment (p<0.001; **Fig 1H**). Conversely, an abnormal increase in MDA of AS compared to the CON group was notably decreased with MET supplementation (p<0.01; **Fig 1I**).

### MET ameliorated atherosclerotic pathological lesion

To investigate the effects of MET treatment on the atherosclerotic lesion in AS, pathological staining including face oil red O staining, oil red O staining, Masson's trichrome staining, and HE staining were used to measure atherosclerotic plaque, fibrosis, and pathological damage in the aortic root of heart, respectively (**Fig 2A**). The percentage of face oil red O staining in the AS group was notably higher than that in the CON group (p<0.001; **Fig 2B**). Similar aggregated results of oil red O staining (p<0.001) and Masson's trichrome staining (p<0.001) were separately observed in AS model, compared to the CON group (**Fig 2C and 2D**). Intriguingly, the attenuation of pathological lesions in AS with MET administration was observed in the above pathological detection. Taken together, these results demonstrated that MET intervention could ameliorate the atherosclerotic lesion.

### MET reduced inflammation in AS

Due to the important role of chronic inflammation in the pathogenesis and development of AS [11], the levels of inflammatory cytokines in plasma and aorta root tissue were respectively determined. Plasma TNF-α (p<0.01; **Fig 3A**), IL-6 (p<0.05; **Fig 3B**), IL-1β (p<0.001; **Fig 3C**),

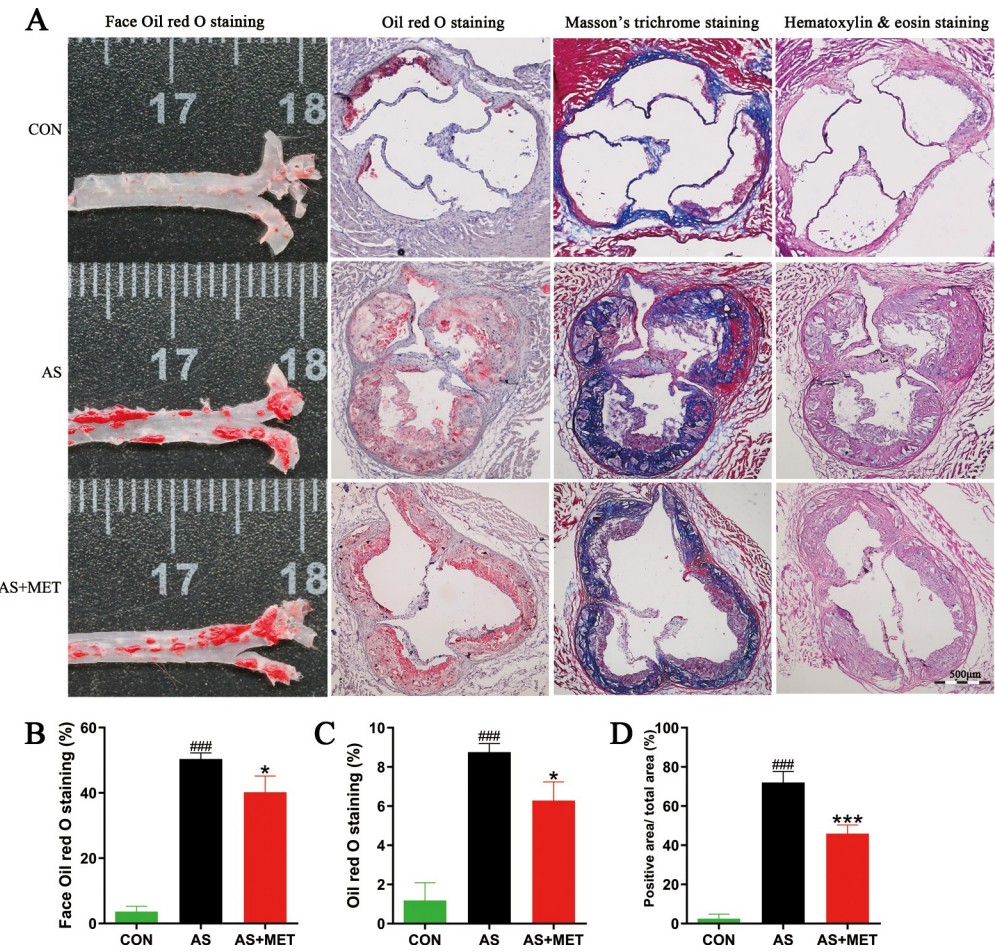

**Fig 2. Effects of MET administration on the pathological lesions in atherosclerotic *ApoE*⁻ᐟ⁻ mice. (A)** Representative sections of the valve area of the aortic root of the heart were stained with face oil red O staining; oil red O staining; Masson's trichrome staining and hematoxylin&eosin staining; respectively. Quantitative analysis as lesion area/total area (%) shown in face oil red O staining (**B**); oil red O staining (**C**); and Masson's trichrome staining (**D**); #(CON vs. AS): *p<0.05, ##p< 0.01, ###p< 0.001. *(AS vs. AS+MET): *p<0.05, **p<0.01, ***p<0.001. Original magnification, ×40. The bar of 500 μm was presented in the right corner of **Fig 2A**.

and IL-10 (p<0.001; **Fig 3D**) in AS group was increased compared with those in the CON group. Importantly, these elevated plasma TNF-α (p<0.001; **Fig 3A**) and IL-6 (p<0.05; **Fig 3B**) in AS could be significantly reduced by MET administration, whereas IL-1β and anti-inflammatory IL-10 concentrations in plasma showed no significant difference (p>0.05; **Fig 3C and 3D**). Similarly, MET administration markedly suppressed aorta inflammation of AS via decreasing aorta pro-inflammatory TNF-α (p<0.05; **Fig 3E**), IL-6 (p<0.05; **Fig 3F**), and IL-1β (p<0.05; **Fig 3G**), but with limited influence in anti-inflammatory IL-10 (p>0.05; **Fig 3H**).

## MET reduced endotoxemia in AS

LPS plays a critical role in triggering chronic inflammation in metabolic diseases, translocation of which mainly due to impaired permeability and integrity of the intestinal barrier [32]. Plasma LPS levels in the diverse groups were determined and found an increase in AS model compared to the CON group, which was conversely decreased with the MET treatment,

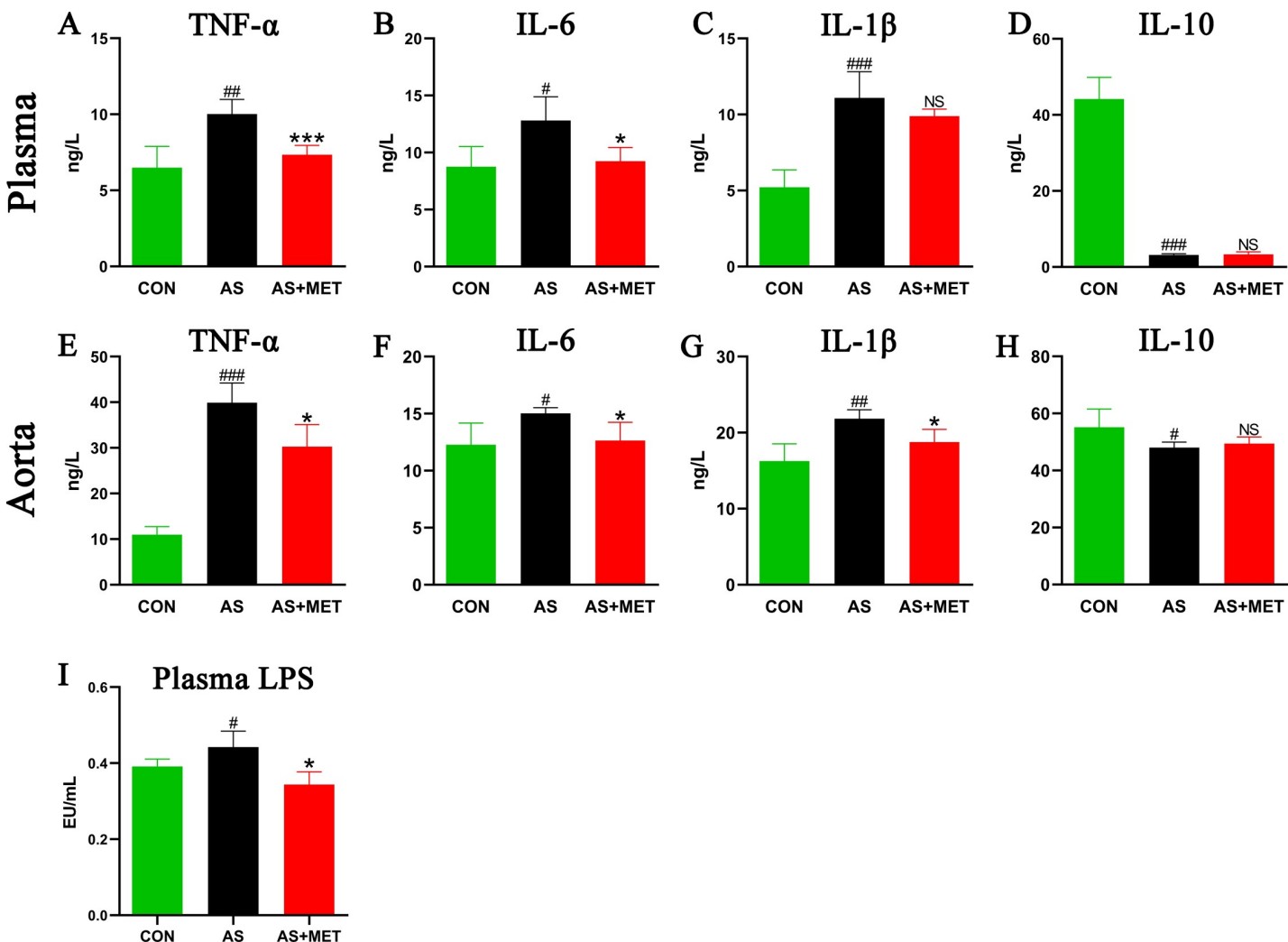

**Fig 3. Determination of plasma and aorta inflammatory cytokine levels in different groups.** Plasma and aorta tissues of mice from three groups were respectively collected for the determination of tumor necrosis factor (TNF)-α (**A, E**); interleukin (IL)-6 (**B, F**); IL-1β (**C, G**) and IL-10 (**D, H**) concentrations using a flow cytometric bead array (CBA) kit. (**I**) Plasma lipopolysaccharide (LPS) levels in diverse groups were determined using a Limulus amebocyte lysate kit. #(CON vs. AS): #p< 0.05, ##p< 0.01, ###p< 0.001. *(AS vs. AS+MET): *p<0.05, **p<0.01, ***p<0.001. NS, no significant difference.

indicating that MET intervention possessed the ability to attenuate gut microbial-derived endotoxemia in atherosclerotic *ApoE⁻/⁻* mice (p<0.05; **Fig 3I**).

## MET reduced atherosclerotic Mψs

Mψs has been solidly proven to play a critical role in the chronic inflammation of AS [33]. Thus, the aorta Mψs were measured by immunofluorescence. We found an increase of F4/80⁺ Mψs in AS (p<0.001; **Fig 4A and 4B**). Moreover, this elevated aorta Mψs in AS was significantly suppressed by MET administration (p<0.01; **Fig 4A and 4B**).

## MET rectified gut dysbiosis in AS

Accumulating reports have addressed the crucial role of the gut microbiome in AS [34]. In the fecal metagenomic analysis of this study, the 16S rRNA sequencing raw reads of gut microbiota in all groups have been submitted in NCBI SRA with an accession number

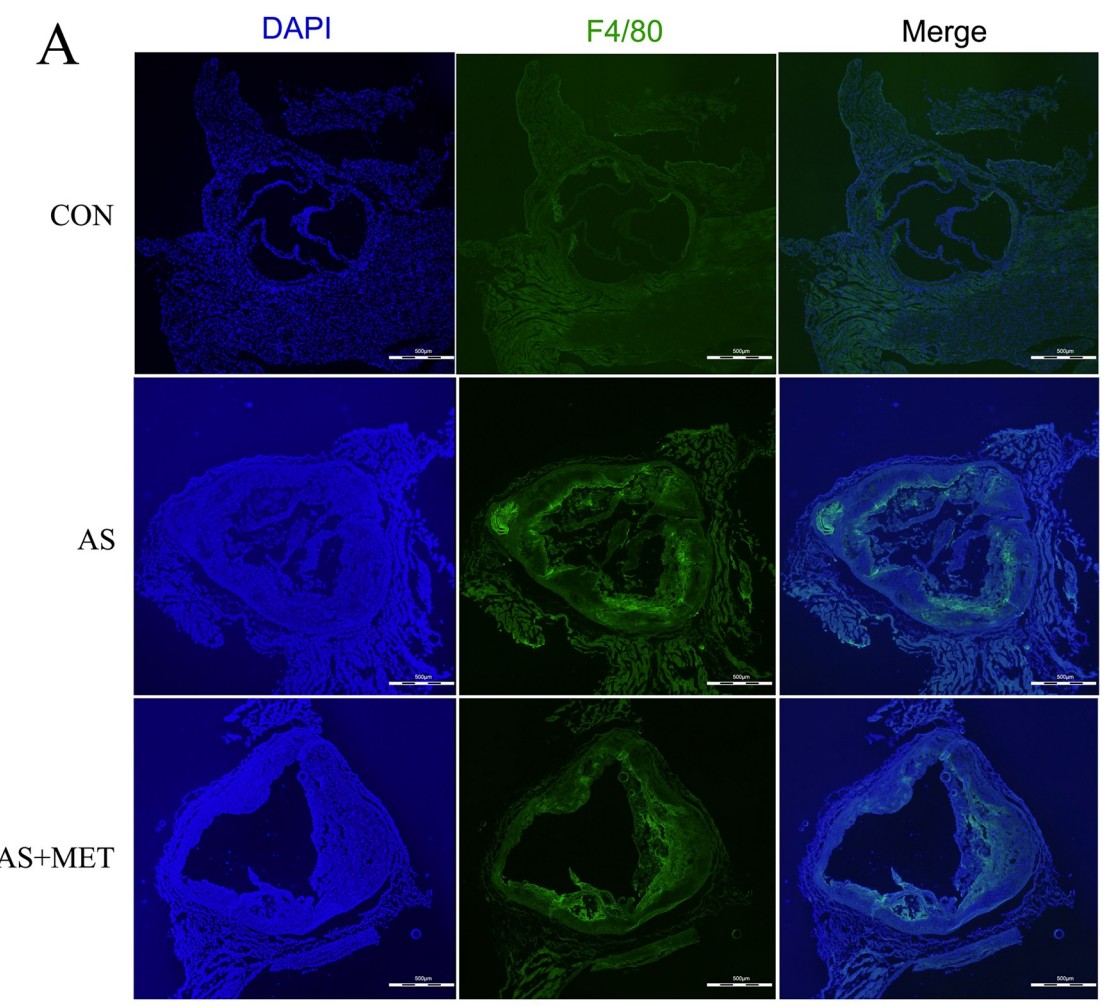

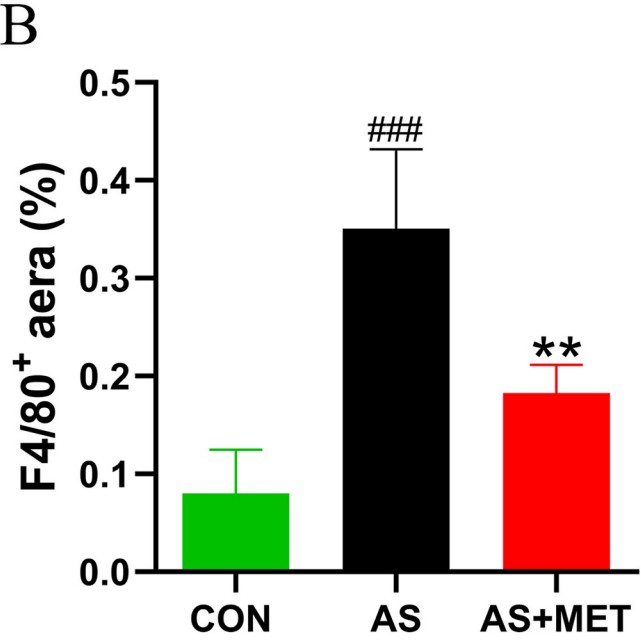

**Fig 4. Determination of cardiac macrophages (Mψs) in diverse groups. (A)** Quantitative analysis as F4/80$^+$ cells/total cells. Original magnification, ×40. The bar of 500 μm was presented in the right corner of the Fig 4A. **(B)** The proportions of F4/80$^+$ cells (Mψs) in diverse groups. $^\#$(CON vs. AS): $^\#$p<0.05, $^{\#\#}$p< 0.01, $^{\#\#\#}$p< 0.001. $^*$(AS vs. AS+MET): $^*$p<0.05, $^{**}$p<0.01, $^{***}$p<0.001.

PRJNA624814. The observed species index displayed significant species diversity between AS model and CON groups (Fig 5A). Rarefaction curves in diverse groups were tended to be flat at 10,000 sequence numbers, indicating that the sequencing data was reasonable (Fig 5B). As β-diversity indicators, PCoA (Fig 5C) and NMDS (Fig 5D) analysis illustrated that the overall bacterial community in the AS group was obviously different from the CON group or AS +MET group. In Venn analysis, 285 core species were observed in all 3 groups, whereas 64, 48, or 30 species were specifically found in CON, AS or AS+MET group (Fig 5G).

To further identify the differential intestinal bacteria in AS with or without MET treatment, we checked the gut microbiota at the phylum level and genus level. We found the relative abundances of *Firmicutes* and *Bacteroidetes* were predominant in all groups at the phylum level (Fig 5E). Predominant *Firmicutes* (p<0.001) and *Proteobacteria* (p<0.01) in AS model were notably increased compared to the CON group, whereas *Bacteroidetes* (p<0.001) was significantly decreased (Fig 5H). Consistently, the ratio of *Firmicutes* to *Bacteroidetes* (F/B ratio) (p<0.001) in AS was dramatically higher than that in CON group (Fig 5J). Intriguingly, *Proteobacteria* (p<0.01) and F/B ratio (p<0.05) were rectified after MET intervention (Fig 5I and 5J), suggesting that oral MET dramatically modulated the gut microbiota at the phylum level in AS.

Moreover, at the genus level (Fig 5F), we found after MET administration, the relative abundances of genera *Akkermansia* (p<0.001; Fig 5K) and *Bifidobacterium* (p<0.01; Fig 5I) were elevated, but *Romboutsia* (p<0.001; Fig 5M) was reduced. Taken together, our genera data indicated that under this experimental condition, MET treatment had a major effect on the microbial community which may contribute to the effectiveness of MET on AS progression.

## MET increased microbial SCFAs

Accumulating studies have suggested that gut microbiota-derived SCFAs as vital microbial metabolites are conductive to regulating the progression of AS [18]. Fecal SCFAs of mice in diverse groups were respectively detected by GC-MS (Fig 6A). The amounts of acetic acid (p<0.001; Fig 6B), propionic acid (p<0.001; Fig 6C), butyric acid (p<0.001; Fig 6D), and valeric acid (p<0.001; Fig 6E) were lower in AS than those in the CON group. However, MET intervention remarkably improved these abnormal SCFAs (Fig 6B–6E), whereas other contents of SCFAs including Isobutyric acid (Fig 6F), Isovaleric acid (Fig 6G), and Caproic acid (Fig 6H) showed no significant alteration (p>0.05), indicating that MET treatment attenuated AS partially via enhancing the generation of crucial SCFAs.

## MET restored the integrity of gut mucosa

To further assess the integrity of gut mucosa after the above rectification of gut dysbiosis with MET treatment, tight junction protein ZO-1 expression level in diverse groups was detected by immunofluorescence staining (Fig 7A). Compared to the CON group, intestinal ZO-1 expression in AS group was significantly reduced, indicating that the integrity of gut mucosa was impaired in AS. Furthermore, gut mucosal ZO-1 level of AS mice showed a notable elevation after the supplementation with MET, demonstrating that MET administration may contribute to enhancing integrity of the gut barrier (p<0.05; Fig 7B). Moreover, the above

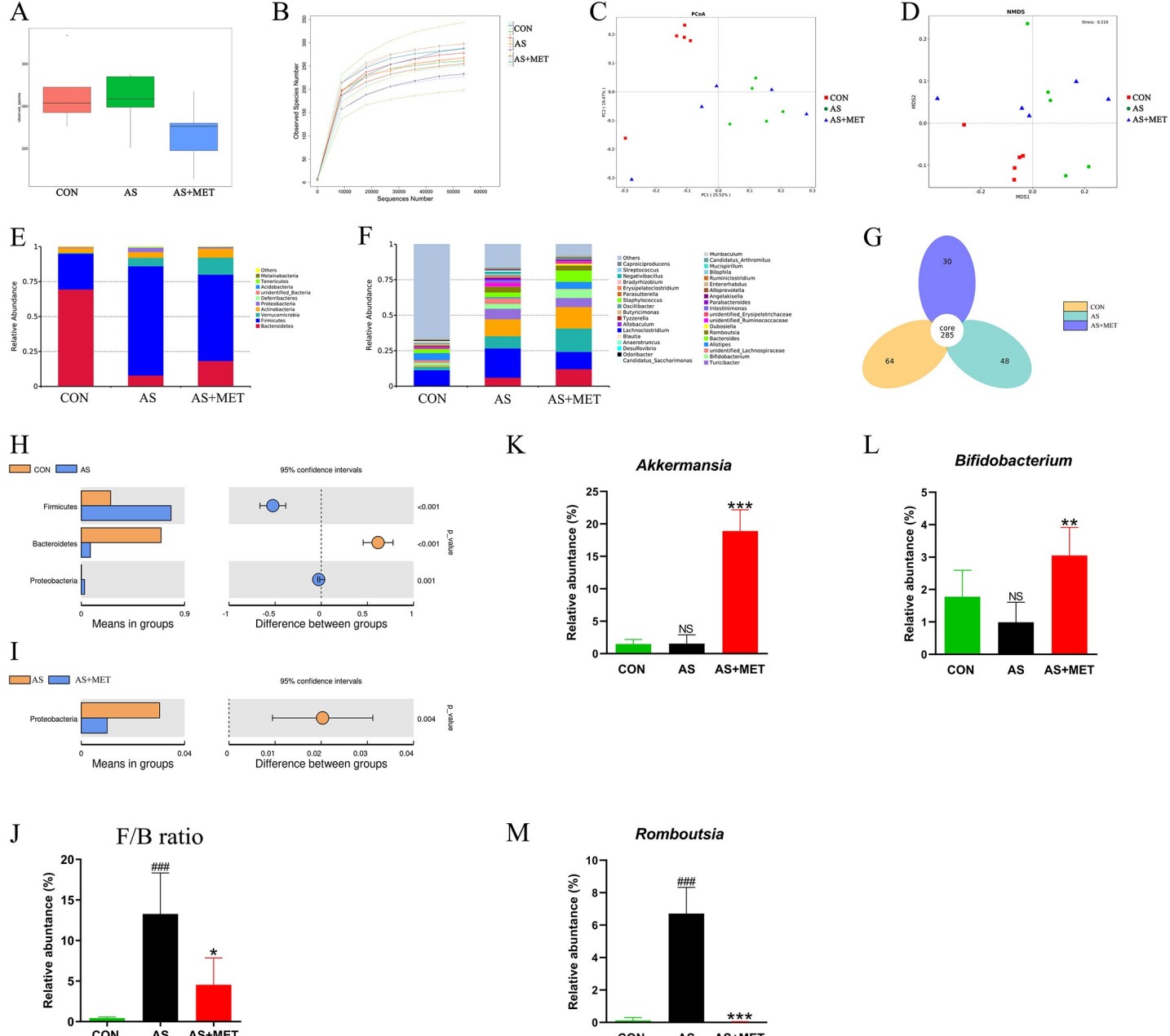

**Fig 5. Gut microbial community in fecal samples of different groups. (A)** Observed-species index; **(B)** Rarefaction Curve; **(C)** PCoA analysis; **(D)** NDMS analysis; **(E, H, I, J)** The phylum level; **(F, K, L, M)** The genus level; **(G)** Venn diagram. #(CON vs. AS): #$p<0.05$, ##$p< 0.01$, ###$p< 0.001$. *(AS vs. AS+MET): *$p<0.05$, **$p<0.01$, ***$p<0.001$. NS, no significant difference.

reduced LPS translocation into plasma also indicated this attenuation of MET on intestinal integrity.

## Correlation analysis

Correlations of the above differential bacteria proportions with inflammatory indicators and SCFAs were analyzed (Fig 8). The abundance of *Firmicutes* was negatively correlated with butyric acid or acetic acid, whereas positively correlated with TNF-α. *Bacteroidetes* was

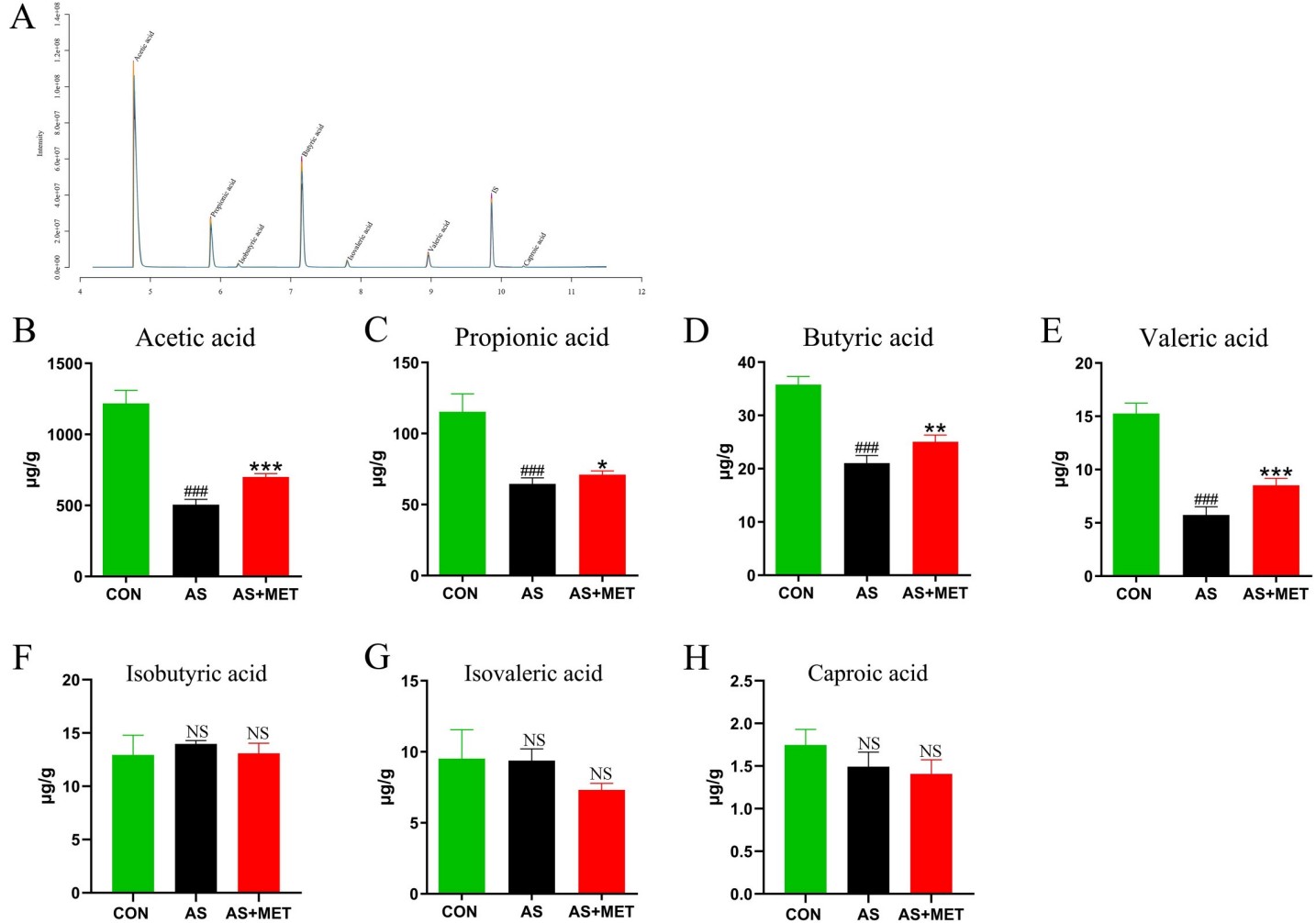

**Fig 6. Measurement of the contents of SCFAs in different groups.** Chromatogram of mice stool (**A**). Determination of levels (μg/g) of Acetic acid (**B**); Propionic acid (**C**); Butyric acid (**D**); Valeric acid (**E**); Isobutyric acid (**F**); Isovaleric acid (**G**) and Caproic acid (**H**) by gas chromatography-mass spectrometer (GC-MS). #(CON vs. AS): #$p < 0.05$, ##$p < 0.01$, ###$p < 0.001$. *(AS vs. AS+MET): *$p < 0.05$, **$p < 0.01$, ***$p < 0.001$. NS, no significant difference.

negatively correlated with plasma TNF-α, aorta IL-1β or IL-6, but positively associated with acetic acid or butyric acid, respectively. Moreover, the ratio of F/B was positively correlated with plasma TNF-α, IL-6 as well as aorta TNF-α and IL-1β, whereas negatively correlated with acetic acid and butyric acid, respectively. Moreover, the relative abundance of *Proteobacteria* was positively correlated with plasma TNF-α, IL-6 and aorta IL-6, but negatively correlated with acetic acid. Genera *Akkermansia* was found to be negatively related to plasma LPS and aorta IL-6 while positively correlated with propionic acid. *Bifidobacterium* was negatively connected with plasma LPS, TNF-α and aorta IL-1β, respectively. Additionally, *Romboutsia* was positively correlated with serum and aorta plasma and aorta inflammation indicators (IL-6, IL-1β and TNF-α), but negatively correlated with differential SCFAs respectively (**Fig 8A**).

Further correlations between SCFAs and inflammation indicators (**Fig 8B**) were analyzed. Acetic acid was negatively correlated with all inflammation indicators, respectively. Similarly, propionic acid was negatively correlated with plasma LPS and aorta IL-1β. Moreover, butyric acid was conversely associated with inflammatory indicators (including plasma TNF-α and IL-6, as well as aorta TNF-α). Additionally, valeric acid was negatively correlated with plasma

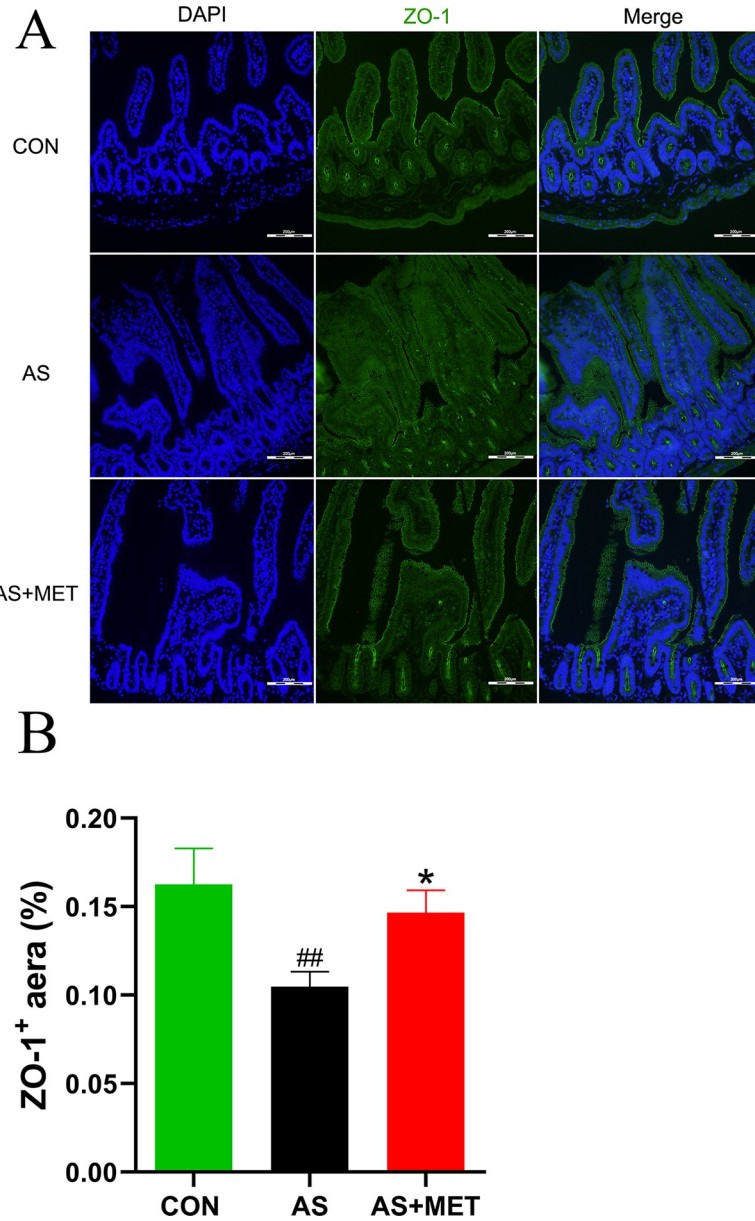

**Fig 7. Determination of intestinal ZO-1 in diverse groups. (A)** Quantitative analysis as ZO-1 area/total area Original magnification, ×100. The bar of 200 μm was presented in the right corner of figure. **(B)** The expression level of ZO-1 in different groups. #(CON vs. AS): #p<0.05, ##p< 0.01, ###p< 0.001. *(AS vs. AS+MET): *p<0.05, **p<0.01, ***p<0.001. NS, no significant difference.

LPS, TNF-α, as well as aorta TNF-α, IL-1β, IL-6 respectively. Thus, gut microbial metabolites SCFAs showed negative correlations with plasma/aorta inflammation.

## Discussion

Treatment with MET has been widely proved to show pleiotropic benefits in diabetes, obesity, and other metabolic disorders [35]. In the present study, we investigated the efficacy of oral MET intervention on AS through regulating gut dysbiosis and inflammation in atherosclerotic *ApoE⁻/⁻* mice. By *in vivo* 12 weeks of intervention, we demonstrated that MET could ameliorate

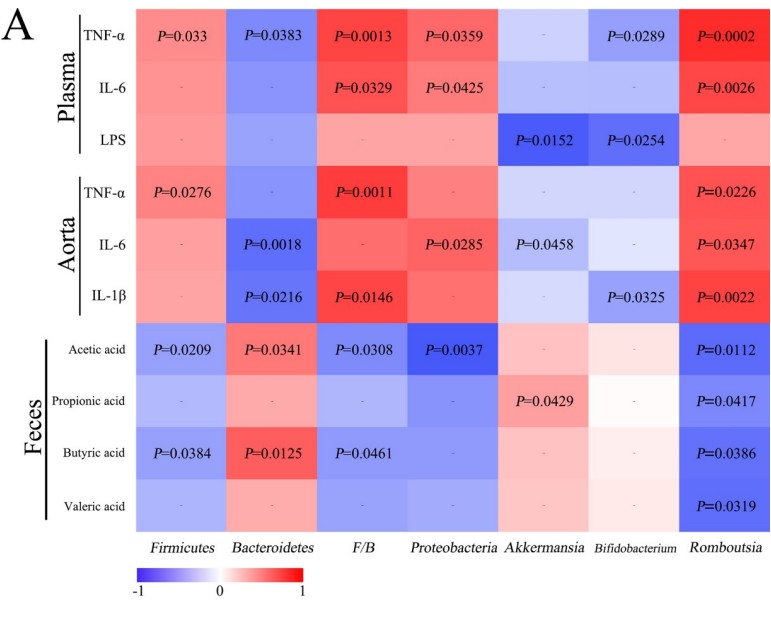

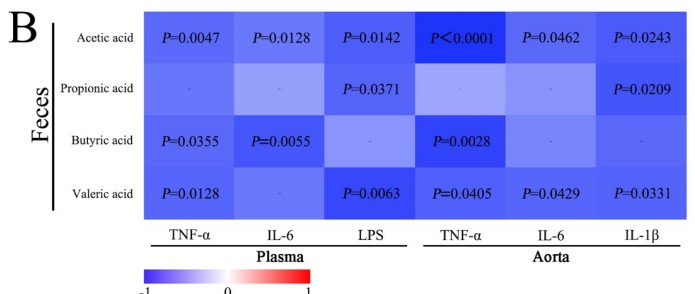

**Fig 8. Correlation analysis. (A)** Correlations of plasma/aorta inflammatory indicators or SCFAs with relative abundance (%) of gut microbiota; **(B)** Correlations between SCFAs and inflammatory indicators.

pathological atherosclerotic lesions and representative indicators of lipid metabolism or oxidative stress. We further revealed that the effectiveness of MET intervention was mainly due to the suppression of inflammation and the modulation of gut microecology.

*ApoE$^{-/-}$* mice characterized by severe hypercholesterolemia due to a virtually completely blocked LDL-receptor mediated lipoprotein remnant clearance, fed with HFD, have been widely reported as a classical AS model [36, 37]. In our study, pathological examinations successfully identified the classical AS model in *ApoE$^{-/-}$* mice. Moreover, MET intervention attenuated the atherosclerotic lesion, demonstrating that long-term MET administration could ameliorate AS in parrallel with previous studies [38, 39].

In this study, a significant reduction of BWs in AS with MET intervention demonstrated that oral MET administration may regulate energy metabolism in AS. Sadeghi et al have shown that MET therapy reduces BMI in adolescents and adults [40]. In addition, no significant change in food intake among diverse groups indicated that the effectiveness of MET intervention was not attributed to the differential food intake.

Hypercholesterolemia is an important risk factor for atherosclerotic CVDs [41]. MET has been thought to show a beneficial effect on the attenuation of lipid metabolism in diverse chronic diseases. MET can significantly ameliorate dyslipidemia including high levels of LDL and TC in type 2 diabetes mellitus (T2DM) [42]. Besides, these results were supported by a

meta-analysis with increased HDL levels after MET treatment [43]. In consistence with previous findings [44], we found the decreases of TG, LDL, TC levels but an increase of HDL after MET treatment, suggesting that MET administration could potently improve atherosclerosis-associated dyslipidemia in AS.

Oxidative stress has been increasingly demonstrated to be involved in the pathophysiological processes of AS [45]. T-SOD and MDA are two representative indicators of oxidative stress [46]. SOD, an antioxidant compound in the endogenous defense system, maintains redox homeostasis by antioxidant reaction [47]. MDA, one of the main products of lipid peroxidation, exists toxic molecule for oxidizing LDL to ox-LDL (oxidized-LDL), eventually leading to the pathogenesis of AS [48]. In this study, increased T-SOD level and decreased MDA level in AS with MET administration indicated that the alleviation of oxidative stress injury may contribute to the effectiveness of MET treatment in anti-atherosclerogenesis, which was in consistence with previous studies [23, 49–51].

Accumulating evidence has proven that the gut microbiome is critical in the occurrence and progress of obesity, diabetes, and cardiovascular disorders [34]. To further understand the mechanism underlying the effectiveness of MET on AS, differential gut microbiota community and associated metabolite SCFAs in diverse groups were investigated. In the present study, decreases in predominant *Firmicutes*, the ratio of *Firmicutes* to *Bacteroidetes*, *Proteobacteria*, *Romboutsia*, as well as increases in *Bacteroidetes*, *Akkermansia*, *Bifidobacterium* with MET treatment suggested that the effectiveness of MET treatment in anti-atherosclerogenes may largely attribute to the rectification of gut dysbiosis, which was paralleled with previous studies [27, 52, 53]. In patients with coronary artery disease (CAD), the ratio of *Firmicutes* to *Bacteroidetes* was increased [54]. A rise in the *Firmicutes*/*Bacteroidetes* ratio was related to an increase in inflammation and an increased capability of harvesting energy from food [55–57]. A study showed a significant decrease of *Akkermansia* in atherosclerotic *ApoE*⁻/⁻ mice [58]. MET has been proven to improve metabolic profiles in aged obese mice partially through modulating gut *Akkermansia* [40]. Transplantation of extracellular vesicles of *Akkermansia muciniphila* improved the BW and lipid profiles of the mice [59]. MET is the most frequently administered medication to treat patients with T2DM via increasing mucin-degrading *Akkermansia* [60]. Increased level of the genera *Bifidobacterium* contributes to the attenuation of tri-methylamine-N-oxide (TMAO)-induced AS [61]. To reveal the key functions and specific mechanisms of differential bacteria in AS progression with MET intervention, high-throughput shotgun metagenomics will be further investigated in our ongoing research.

Based on gut dysbiosis, LPS is thought to be critical in the inflammation of AS [62]. LPS, a causal link between gut microbiota and systemic inflammation, generated from pathogenic bacteria, translocates to the artery and then binds to TLR-4 of Mψs to induce an inflammatory cascade reaction to lead to AS ultimately [63]. Endotoxemia evokes the polarization and activation of inflammatory macrophage (M1) and promotes foam cell formation in AS. Intriguingly, in this study, the decrease of plasma LPS level with MET administration revealed that the potent anti-inflammatory effect of MET may probably depend on the LPS-mediated inflammatory pathway. Furthermore, MET can attenuate endotoxemia in chronic metabolic diseases by regulating the gut microbiota [64]. Critical role of LPS-TLR4 mediated inflammatory pathway and associated polarization of Mψs in alcoholic liver disease have been previously demonstrated by our lab [28, 65].

Mψs are thought to the crucial inflammatory cells in the pathogenesis and progression of AS [66]. For the current study, an increase of aorta Mψs in AS indicated that inflammatory Mψs mediated inflammation may aggravate the severity of the disease via elevated LPS-TLR4 signaling. Importantly, MET intervention attenuated the number of infiltrated Mψs in the aorta of AS, suggesting the amelioration of AS with MET treatment may due to macrophage-mediated

inflammation. Ge Tang et al have found MET can inhibit NLRP3 inflammasomes activation and suppress diabetes-accelerated atherosclerosis in *ApoE*$^{-/-}$ mice [38]. However, the activation and proliferation of Mψs, as well as the inflammatory axis in macrophage polarization (M1/M2) in MET treatment on AS need further investigated.

Consistent with decreased plasma LPS, as representative pro-inflammatory indicators, IL-6 and TNF-α levels in plasma were decreased with MET intervention, demonstrating that MET may alleviate AS via suppressing the chronic inflammation. However, we speculate that the limited effect of MET on anti-inflammatory IL-10 may probably due to a complicated role in modulating pro- and anti-inflammation in AS. A similar study reported that combined inflammatory cytokines led to a chronic inflammatory response in the vessel wall, which was thought to be responsible for disease progression characterized by a buildup of atherosclerotic plaque. Moreover, the above significant alteration of Mψs may be the primary source of these pro-inflammatory cytokines through LPS-TLR-NF-κB/Nod-like receptor protein 3 (NLRP3) inflammasomes signaling [67]. However, whether other immune cells such as regulatory T cells (Tregs), Th17 cells and Myeloid suppressor cells (MDSCs) are involved in the anti-inflammatory effects of the MET treatment on AS still needs to be further explored.

SCFAs with less than six carbons are the important end-products of gut microbiota metabolites after diet digestion and fermentation, linking interaction between gut microbiota and host homeostasis in the regulation of inflammation and metabolism [18]. SCFAs exhibit a wide range of physiological functions including histone deacetylases inhibition [68], chemotaxis and phagocytosis modulation [69], reactive oxygen species induction [70], cell proliferation [71], and intestinal barrier integrity alteration [72]. Predominant acetate, propionate and butyrate were composed of 90% of SCFAs. Acetate and propionate are mainly produced by *Bacteroidetes* fermentation. Butyrate is the main product of *Firmicutes*. Studieshave shown that administration of *Akkermansia* improves metabolic phenotypes in mice [59]. Intriguingly, MET can increase the relative abundance of *Akkermansia* [73]. Napolitano *et al.* identified the changes in the relative concentration of phyla *Bacteroides* and *Firmicutes* with MET treatment [74]. Several other studies also demonstrated that reprogramming of gut microbiota or transplantation of fecal microbiota could modulate the SCFAs to prevent obesity and cerebral ischemic stroke [75, 76]. In addition to the gut microbiota, dietary composition also influences the production of SCFAs and subsequent functional readout. Ryan et al. demonstrated the shifts in the composition of the gut microbiome in *ApoE*$^{-/-}$ mice fed high fat/cholesterol in conjunction with plant sterol esters or oat β-glucan could lead to increased concentrations of cecum acetate and butyrate to contribute to the reduction of serum cholesterol concentrations [77]. In our tudy, MET administration can restore the abnormal decreased predominant contents of SCFAs, revealing that MET administration may ameliorate AS partially via modulating gut microbial SCFAs. Besides, we also observed that increased SCFAs-producing *Bacteroidetes*, decreased *Firmicutes*, and reduced the F/B ratio, which was consistent with previous studies [78–80]. MET treatment in mice was found to modulate gut microbiota and increase SCFA-metabolizing bacteria [80]. In our study, we found a positive correlation between SCFAs and *Akkermansia*, which was consistent with a cohort study [60]. MET shifts gut microbiota composition through the enrichment of mucin-degrading *A.muciniphila* as well as several SCFA-producing microbiota. SCFAs served as one of the energy resources that play a critical role in the cardiovascular system [81]. In this study, we consider the impact of notable increases of SCFAs with MET administration on the attenuation of AS may probably rely on the suppression of inflammation via binding to G-protein coupled receptor (GPR)41 or GPR43 receptors on Mψs [82]. Recent studies also strongly suggested that the microbial SCFAs inhibited the inflammation through regulating the activity of Histone Deacetylase (HDAC) [83].

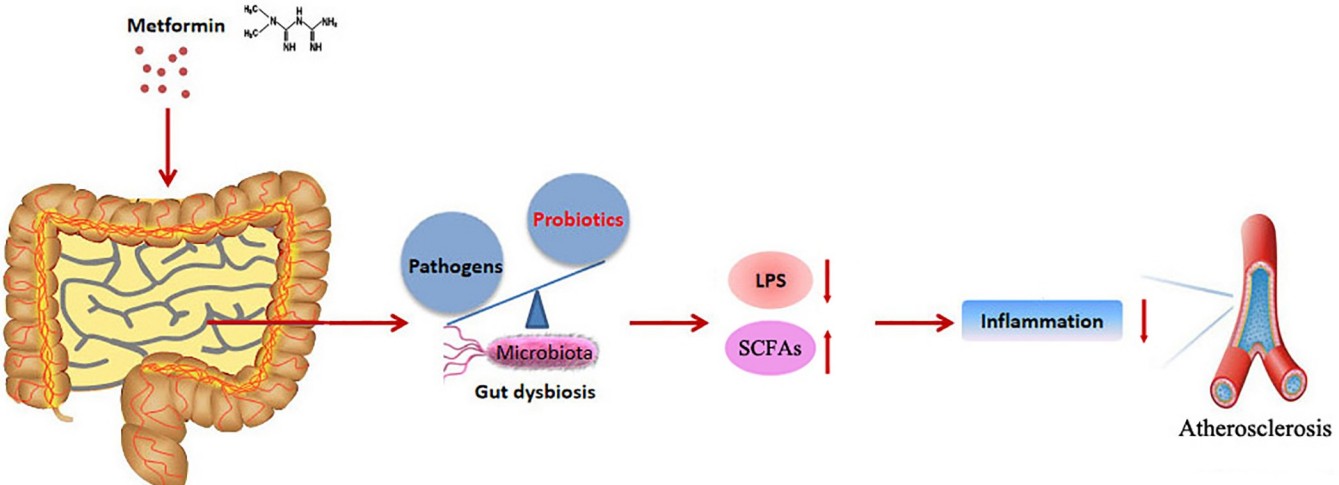

**Fig 9. Patterns of the amelioration of metformin (MET) in atherosclerosis (AS) through restoring gut dysbiosis and suppressing inflammation in atherosclerotic $ApoE^{-/-}$ mice.**

Nevertheless, the exact mechanism of SCFAs in the effectiveness of MET treatment needs to be further investigated.

The protective role of intestinal tight junction proteins in the integrity and permeability of gut microecology has been solidly demonstrated in chronic metabolic diseases [84, 85]. In our study, in parallel with the rectification of gut dysbiosis and the suppression of inflammation after MET intervention, the improvement of tight junction protein ZO-1 revealed that oral MET intervention may contribute to the attenuation of the integrity of gut barrier in AS. The improvement of impaired intestinal integrity may improve the permeability of the gut barrier and subsequently reduce LPS translocation, ultimately suppress atherosclerotic chronic low-degree inflammation.

## Conclusion

Our study highlighted that MET treatment ameliorated AS progression through anti-inflammation and restoring gut dysbiosis in atherosclerotic $ApoE^{-/-}$ mice, which could contribute to the understanding of the underlying mechanism of MET in AS treatment and potentially promote MET served as an inexpensive and effective intervention for the control of the atherosclerotic cardiovascular disease. The schematic diagram was illustrated in **Fig 9**.

## Acknowledgments

The authors thank Prof. Xiaoxia Zhang and Prof. Jun He for their skillful technical assistance.

## Author Contributions

**Conceptualization:** Ning Yan, Libo Yang, Hao Wang, Shaobin Jia.

**Data curation:** Ning Yan, Yiwei Li, Ting Wang, Hao Wang, Shaobin Jia.

**Formal analysis:** Lijuan Wang, Ting Wang, Ru Yan.

**Funding acquisition:** Hao Wang, Shaobin Jia.

**Investigation:** Ning Yan, Lijuan Wang, Yiwei Li, Ru Yan.

**Methodology:** Ning Yan, Lijuan Wang, Yiwei Li, Ru Yan.

**Project administration:** Ning Yan.

**Software:** Lijuan Wang, Ting Wang, Libo Yang.

**Supervision:** Hao Wang, Shaobin Jia.

**Visualization:** Lijuan Wang, Ting Wang.

**Writing – original draft:** Ning Yan, Lijuan Wang, Yiwei Li.

**Writing – review & editing:** Hao Wang, Shaobin Jia.

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
