## [Decision Letter · Decision Letter 0]

21 Apr 2021

PONE-D-21-07383

M etformin intervention ameliorates AS in ApoE-/- mice through restoring gut dysbiosis and anti-inflammation

PLOS ONE

Dear Dr. Jia,

Thank you for submitting your manuscript to PLOS ONE. After careful consideration, we feel that it has merit but does not fully meet PLOS ONE’s publication criteria as it currently stands. Therefore, we invite you to submit a revised version of the manuscript that addresses the points raised during the review process.

We look forward to receiving your revised manuscript.

Kind regards,

Michael Bader

Academic Editor

PLOS ONE

Journal Requirements:

2. To comply with PLOS ONE submissions requirements, please provide methods of sacrifice in the Methods section of your manuscript.

[This work was supported by National Natural Science Foundation, China (Grant number 82060057); Key Research and Development Projects ofNingxia, China (Grant number 2018BEG02006);Nat-ural Science Foundation of Ningxia, China (Grant number 2018AAC03257); Program for Con-structing Superior Subject-groups of Cardiovascular Disease in Ningxia Medical University, Ning-xia, China (Grant number 2001210501);Ningxia High School First-class Disciplines (Basic Medical Sciences inNingxia Medical University), China (Grant number NXYLXK2017B07).]

 [The funders had no role in study design, data collection and analysis, decision to publish, or preparation of the manuscript.]

5. We noticed you have some minor occurrence of overlapping text with the following previous publication(s), which needs to be addressed:

- https://portlandpress.com/clinsci/article-abstract/134/6/657/222480/Short-chain-fatty-acid-acylation-and?redirectedFrom=fulltext

The text that needs to be addressed involves page 12, paragraph 3.

In your revision ensure you cite all your sources (including your own works), and quote or rephrase any duplicated text outside the methods section. Further consideration is dependent on these concerns being addressed.

Reviewers' comments:

Reviewer's Responses to Questions

**Comments to the Author**

1. Is the manuscript technically sound, and do the data support the conclusions?

Reviewer #1: Partly

Reviewer #2: Yes

2. Has the statistical analysis been performed appropriately and rigorously? 

Reviewer #1: I Don't Know

Reviewer #2: Yes

3. Have the authors made all data underlying the findings in their manuscript fully available?

Reviewer #1: Yes

Reviewer #2: Yes

4. Is the manuscript presented in an intelligible fashion and written in standard English?

Reviewer #1: Yes

Reviewer #2: Yes

5. Review Comments to the Author

Reviewer #1: Yan et al. conducted an ApoE (-/-) mice study to investigate the pleiotropic effects of metformin on atherosclerosis protection including potential microbiome-related pathways. The authors demonstrated the intervention of metformin improved atherosclerotic phenotypes in ApoE (-/-) mice, inflammatory biomarkers and gut dysbiosis (including increase in beneficial commensal gut bacteria and SCFAs). Although the authors did not conduct a germ-free mice study to prove the cause-effective relationship between the microbiome shift and improvement of atherosclerosis, significant correlations between specific bacteria/SCFA and the inflammatory markers at atherosclerotic plaques were demonstrated in the paper.

My comments are as follows.

1. According to Fig 7a., the expression of ZO-1 seems to be increased in AS group as compared with CON group while the AS+MET has similar ZO-1 expression with CON. However, in Fig 7b, the bar plot showed ZO-1 area was significantly decreased in AS group as compared with CON group while the AS+MET significantly restored the ZO-1 area when compared with AAS group. The Fig7a and Fig7b seems to show contradictory results, please explain it.

2. The contradictory results showed in Fig7 were also reflected in the manuscript. In the abstract, the sentence “…..attenuation of gut dysbiosis, intestinal tight junction protein ZO-1 in AS was elevated.” suggests the increased expression of tight junction protein in AS group. However, the author also showed increased endotoxemia (Fig 3i) in AS group. It’s weird that an enhanced intestinal integrity and a sign of leaky gut (increased LPS) coexist at the same time. Please explain it.

3. Please also, recheck the rationale of the whole paragraph of “MET attenuated the integrity of gut mucosa”. Many contradictions were observed.

4. Please reupload the Fig 5 with a higher resolution. The data and figures showed here are too blurred to be interpretated.

5. In Fig 9, the red arrow indicated promotion. However, the picture here may be misleading because it looks like the gut dysbiosis will “promote” a decrease in LPS and an increase in SCFAs. Please revise it.

Reviewer #2: The manuscript by Ning Yan et al. examines the effect of metformin intervention on atherosclerosis and the role of the microbiome in this process using a genetic mouse model. This is a well-performed study that contains information on multiple parameters and provides new additional information on metformin as a pleiotropic drug in correlation with AS. However, some parts of the manuscript require further clarification.

Specific comments:

One of the limitations of the presented study is the useof the 16S rRNA gene sequencing as the choice of method, and it would be advisable to perform shotgun metagenomic sequencing. 16S rRNA sequencing allows determining microbiota to the genus level only, which may not be sufficient to explain the associations. Also, shotgun metagenome would provide a realistic view of the microbiome's functional profiling. This should be mentioned as a limitation.

A significant problem investigating the microbiome in mice studies is the interindividual variation in the content of microbial species. One of the prerequisites for such studies is to monitor baseline microbiome composition before the intervention. This would allow to understand if differences in microbiota between different groups are not mainly determined by baseline variation in the microbiome. From this aspect, longitudinal sampling would be beneficial. In any case, a more detailed description of the study is needed (for example, explaining if all the animals came from the same batch), including the conditions of housing and precautions to avoid contamination of animals with environmental microbiota during the experiment.

From the description of the methods, it is unclear which technology has been used for sequencing: Ion S5 TM XL platform or Illumina HiSeq 2500 platform. This should be clarified. Also, details on bioinformatic analysis and description of NGS data quality (e.g., number of reads per sample) are not sufficient.

16S rRNA gene sequencing is well known to be very sensitive to contamination. Authors should describe procedures applied to minimize this risk (for example, if sequencing of negative control was performed).

The authors have to improve the legend of figure 5 to verify its conformity with actual parts of the figure and make it more clear to the reader.

6. PLOS authors have the option to publish the peer review history of their article (what does this mean?). If published, this will include your full peer review and any attached files.

Reviewer #1: **Yes: **Wei-Kai Wu

Reviewer #2: No

---

## [Author Response · Author response to Decision Letter 0]

2 Jun 2021

Dear editor,

Thank you for your kind letter. Thanks for the reviewers’ valuable comments on our manuscript. We have revised the manuscript in accordance with the comments, and carefully proofread the manuscript to minimize typographical, grammatical, and bibliographical errors.

We have uploaded 4 files (Response to Reviewers, Revised Manuscript with Track Changes, Manuscript, and Cover Letter) to the submission system, and removed the Funding-related information from the manuscript to the Funding Information section of the submission system. All alterations in manuscript were marked with red color. 

Response to editor comments 

According to all of your kind comments, our manuscript and submission have been carefully revised, please check them.

1. Please ensure that your manuscript meets PLOS ONE's style requirements, including those for file naming. The PLOS ONE style templates can be found at https://journals.plos.org/plosone/s/file?id=wjVg/PLOSOne_formatting_sample_main_body.pdf and https://journals.plos.org/plosone/s/file?id=ba62/PLOSOne_formatting_sample_title_authors_affiliations.pdf.

Reply: We really appreciate your careful comments. According to the PLOS ONE style templates provided in the above links, we have immediately revised the file naming, as well as other associated styles, to make sure our manuscript meets your PLOS ONE's style requirements.

2. To comply with PLOS ONE submissions requirements, please provide methods of sacrifice in the Methods section of your manuscript.

Reply: Thank you for your comments. According to your helpful advice, the methods of sacrifice were immediately added in the Methods section as“After 12 weeks of intervention, all the mice were euthanized by 4% sodium pentobarbital and samples were collected”.

Reply: Thank you for your kind advice. According to your suggestion, the manuscript have been thoroughly revised to minimize typographical, grammatical, and bibliographical errors. Please check changes with red color in the Revised Manuscript with Track Changes.

[This work was supported by National Natural Science Foundation, China (Grant number 82060057); Key Research and Development Projects of Ningxia, China (Grant number 2018BEG02006);Natural Science Foundation of Ningxia, China (Grant number 2018AAC03257); Program for Con-structing Superior Subject-groups of Cardiovascular Disease in Ningxia Medical University, Ning-xia, China (Grant number 2001210501);Ningxia High School First-class Disciplines (Basic Medical Sciences in Ningxia Medical University), China (Grant number NXYLXK2017B07).] We note that you have provided funding information that is not currently declared in your Funding Statement. However, funding information should not appear in the Acknowledgments section or other areas of your manuscript. We will only publish funding information present in the Funding Statement section of the online submission form. Please remove any funding-related text from the manuscript and let us know how you would like to update your Funding Statement. Currently, your Funding Statement reads as follows: [The funders had no role in study design, data collection and analysis, decision to publish, or preparation of the manuscript.] Please include your amended statements within your cover letter; we will change the online submission form on your behalf.

Reply: We really appreciate your careful comments. We have deleted any funding-related text from our manuscripts, and uploaded those funding-related information in Funding Information section of the submission system. Please check it, thank you. Our amended statements about the Funding were also involved in our update cover letter as your suggestion.

5.We noticed you have some minor occurrence of overlapping text with the following previous publication(s), which needs to be addressed: - https://portlandpress.com/clinsci/article-abstract/134/6/657/222480/Short-chain-fatty-acid-acylation-and?redirectedFrom=fulltext The text that needs to be addressed involves page 12, paragraph 3.In your revision ensure you cite all your sources (including your own works) and quote or rephrase any duplicated text outside the methods section. A further consideration is dependent on these concerns being addressed 

Reply: Thanks for your careful suggestion. According to your advice, we have immediately checked and added references about all sources (including our own works) and quote or rephrase any duplicated text outside the methods section to ensure the cited references in the manuscript is complete, including the above mentioned reference (https://portlandpress.com/clinsci/article-abstract/134/6/657/222480/Short-chain-fatty-acid-acylation-and?redirectedFrom=fulltext ). Associated changes were marked with red color. 

Response to Reviewer #1

1)According to Fig 7a., the expression of ZO-1 seems to be increased in AS group as compared with the CON group while the AS+MET has a similar ZO-1 expression with CON. However, in Fig 7b, the bar plot showed ZO-1 area was significantly decreased in AS group as compared with CON group while the AS+MET significantly restored the ZO-1 area when compared with AS group. The Fig7a and Fig7b seems to show contradictory results, please explain it.

Reply: We really appreciate for your kind comments. Zonula occludens-1 (ZO-1) protein, an epithelial tight junction protein, was a biomarker of intestinal permeability. With accelerating progression of atherosclerosis, the expression of ZO-1 showed a notable decrease. When drug intervention was used to inhibit the progress of atherosclerosis, the expression of ZO-1 was increased. We found the same trend as Kentaro Arakawa et al study[1]. In our study, in the Fig 7A and Fig 7B, compared with the CON group, the ZO-1 expression in AS group was decreased, and it could be restored by MET intervention. We’re sure there’s no controversial results in ZO-1 expression. Please check the Fig 7 and context in the manuscript, thank you very much.

Reference: [1] Arakawa K, Ishigami T, Nakai-Sugiyama M, et al. Lubiprostone as a potential therapeutic agent to improve intestinal permeability and prevent the development of atherosclerosis in apolipoprotein E-deficient mice[J]. PloS one, 2019, 14(6): e0218096.

2) The contradictory results showed in Fig7 were also reflected in the manuscript. In the abstract, the sentence “…..attenuation of gut dysbiosis, intestinal tight junction protein ZO-1 in AS was elevated.” suggests the increased expression of tight junction protein in AS group. However, the author also showed increased endotoxemia (Fig 3I) in AS group. It’s weird that an enhanced intestinal integrity and a sign of leaky gut (increased LPS) coexist at the same time. Please explain it.

Reply: We really appreciate for your kind comments. Our manuscript is intended to show that MET treatment displayed beneficial effects on the amelioration of AS progression in atherosclerotic ApoE-/- mice through restoring gut dysbiosis, improving the intestinal mucosal barrier, and reducing the systemic inflammation which could contribute to the understanding of the underlying mechanism of MET in AS treatment and potentially promote MET served as an inexpensive and effective intervention for the control of the atherosclerotic cardiovascular disease. In the abstract and Fig 7 and associated context in our manuscript, we described as “ MET attenuation of gut dysbiosis, intestinal tight junction protein ZO-1 in AS was elevated”, it means ZO-1 level is elevated in AS+MET treated group, it does not mean the ZO-1 expression is elevated in AS group. We’re worried that it’s a misunderstanding for the reviewer, please check our manuscript, please do not hesitate to contact with us immediately if necessary. 

3) Please also, recheck the rationale of the whole paragraph of “MET attenuated the integrity of gut mucosa”. Many contradictions were observed. 

Reply: We really appreciate for your careful comments. Questions 1) to 3) are the similar questions. Indeed, about Fig 7 of the result, description sentences have been provided as “Compared to the CON group, intestinal ZO-1 expression in AS group was significantly reduced, indicating that the integrity of gut mucosa was impaired in AS. Furthermore, after the supplementation with MET, gut mucosal ZO-1 level of AS mice showed a notable elevation, demonstrating that MET administration may contribute to attenuating the integrity of the gut barrier (Fig 7B). Moreover, the above-reduced LPS translocation into plasma also indicated this attenuation of intestinal integrity.”, which revealed that ZO-1 in AS group was decreased, not increased. Thus, the meaning of our this result maybe probably misunderstood by the reviewer. Please check and help us to improve our manuscript. Thus, our study demonstrated that “MET attenuated the integrity of gut mucosa”.

4)Please reupload the Fig 5 with a higher resolution. The data and figures showed here are too blurred to be interpretated.

Reply: Thank you for your helpful advice. In accordance with your helpful suggestion for improving the quality of our manuscript, Fig 5 has been immediately revised with a higher resolution. Additionally, we have carefully checked all figures to make sure the figures involved in our manuscript with high quality.

5) In Fig 9, the red arrow indicated promotion. However, the picture here may be misleading because it looks like the gut dysbiosis will “promote” a decrease in LPS and an increase in SCFAs. Please revise it.

Reply: We really appreciate for your comments. According to your valuable advice, the Fig 9 was immediately revised to ensure the figure without any misleading meaning.

Response to Reviewer #2

1) One of the limitations of the presented study is the use of the 16S rRNA gene sequencing as the choice of method, and it would be advisable to perform shotgun metagenomic sequencing. 16S rRNA sequencing allows determining microbiota to the genus level only, which may not be sufficient to explain the associations. Also, shotgun metagenome would provide a realistic view of the microbiome's functional profiling. This should be mentioned as a limitation.

Reply: We really appreciate your kind suggestion. Up to now, there are two main strategies implemented for the analysis of microbial communities through NGS: shotgun metagenomics and 16S rDNA sequencing[1,2]. Shotgun metagenomics consists of the sequencing of bacterial DNA isolated from the whole microbial community. 16S rDNA sequencing relies on the polymerase chain reaction(PCR) amplification of a specific region in the 16S gene. Shotgun metagenomics requires higher coverage (10–30 million reads) and a more complex downstream data analysis, like the GO, KEGG, and metabolic correlation analysis, and what’s roles of gut microbiota in the disease development can be explained in more detail. While, 16S sequencing is generally believed to be a robust, well-characterized method that yields sufficient information about microbial communities’ composition, a major limitation of this method is that taxa are assigned based on the sequence of only a single region of the bacterial genome. We’re really appreciate your valuable advice, the shotgun metagenomics will be used in our ongoing study to help us to further understand the key functions and specific mechanisms of specific bacteria in AS progression as your valuable suggestion, which was also mentioned in Discussion section.

Reference: [1] Durazzi F, Sala C, Castellani G, et al. Comparison between 16S rRNA and shotgun sequencing data for the taxonomic characterization of the gut microbiota[J]. Scientific reports, 2021, 11(1): 1-10.

[2] Laudadio I, Fulci V, Palone F, et al. Quantitative assessment of shotgun metagenomics and 16S rDNA amplicon sequencing in the study of human gut microbiome[J]. Omics: a journal of integrative biology, 2018, 22(4): 248-254.

2) A significant problem investigating the microbiome in mice studies is the interindividual variation in the content of microbial species. One of the prerequisites for such studies is to monitor baseline microbiome composition before the intervention. This would allow understanding if differences in microbiota between different groups are not mainly determined by baseline variation in the microbiome. From this aspect, longitudinal sampling would be beneficial. In any case, a more detailed description of the study is needed (for example, explaining if all the animals came from the same batch), including the conditions of housing and precautions to avoid contamination of animals with environmental microbiota during the experiment.

Reply: We really appreciate for your kind comments. For your concerns, firstly, in our experiment, the same batch ApoE-/- mice (male) from Vital River Laboratory Animal Technology Co., Ltd., Beijing, China (Product Number: scxk2016-0006) were obtained to avoid the interindividual variation and precautions. Secondly, all of mice were housing in specific pathogen-free (SPF) environment with the same condition in the Experiment Animal Centre of Ningxia Medical University. Thirdly, at the end of our experiment, 5 mice from each group were randomly selected to immediately obtain fresh feces samples with sterile cages and nucleic acid-free 1.5mL EP tubes. All the above measures ensured that the mice were under the same baseline without specific pathogen bacteria contamination. Associated detailed descriptions were added in the manuscript with red color.

3) From the description of the methods, it is unclear which technology has been used for sequencing: Ion S5 TM XL platform or Illumina HiSeq 2500 platform. This should be clarified. Also, details on bioinformatic analysis and description of NGS data quality (e.g., number of reads per sample) are not sufficient. 

Reply: Thank you for your careful advice. We’re so sorry that we have made a mistake. According to your valuable comment, we have checked the methods in our study, and ensured that Illumina HiSeq 2500 platform was used, not Ion S5 TM XL platform, as same as our previous studies[1,2], and the related description sentence was immediately corrected. In addition, all raw sequences have been submitted to the Sequences Read Archive database at the NCBI with an accession number PRJNA624814, in convenience with the reader to check all sequencing information. 

References:[1] Ting Wang, Liping Sha, Yiwei Li, Lili Zhu, Zhen Wang, Ke Li, Haixia Lu, Ting Bao, Li Guo, Xiaoxia Zhang and Hao Wang. Dietary ɑ-linolenic acid (ALA)-rich flaxseed oil exerts beneficial effects on polycystic ovary syndrome through sex steroids hormones-microbiota-inflammation axis in rats. 2020, Frontiers in Endocrinology. 11:284. doi: 10.3389/fendo.2020.00284. 

[2]Haixia Lu, Ping Liu, Xiaoxia Zhang, Ting Bao, Ting Wang, Li Guo, Yiwei Li, Xiaoying Dong, Xiaorong Li, Youping Dong, Liping Sha, Lanjie He, Hao Wang. Inulin and Lycium barbarum polysaccharides ameliorate diabetes by enhancing gut barrier via modulating gut microbiota and activating gut mucosal TLR2+ intraepithelial γδ T cells in rats. Journal of Functional Foods, 2021, 79: 104407. DOI: 10.1016/j.jff.2021.104407.

4) 16S rRNA gene sequencing is well known to be very sensitive to contamination. Authors should describe procedures applied to minimize this risk (for example, if sequencing of negative control was performed).

Reply: Thanks for your kind advice. Consistent with your kind comment, we have conducted and mastered the experimental procedures for the 16sRNA sequencing methods as our previously descriptions[1-5]. In addition, CON group was involved in the whole study to minimize the risk of contamination that may lead to unreliable results. Thank you very much for your careful reminding.

Reference:[1] Wang T, Sha L, Li Y, et al. Dietary α-Linolenic acid-rich flaxseed oil exerts beneficial effects on polycystic ovary syndrome through sex steroid hormones—microbiota—inflammation axis in rats[J]. Frontiers in Endocrinology, 2020, 11: 284.

[2] Zhang X, Wang H, Yin P, et al. Flaxseed oil ameliorates alcoholic liver disease via anti-inflammation and modulating gut microbiota in mice[J]. Lipids in Health and Disease, 2017, 16(1): 1-10.

[3] Guo L, Xiao P, Zhang X, et al. Inulin ameliorates schizophrenia via modulation of the gut microbiota and anti-inflammation in mice[J]. Food & Function, 2021, 12(3): 1156-1175.

[4] Bao T, He F, Zhang X, et al. Inulin Exerts Beneficial Effects on Non-Alcoholic Fatty Liver Disease via Modulating gut Microbiome and Suppressing the Lipopolysaccharide-Toll-Like Receptor 4-Mψ-Nuclear Factor-κB-Nod-Like Receptor Protein 3 Pathway via gut-Liver Axis in Mice[J]. Frontiers in Pharmacology, 2020, 11.

[5] Haixia Lu, Ping Liu, Xiaoxia Zhang, Ting Bao, Ting Wang, Li Guo, Yiwei Li, Xiaoying Dong, Xiaorong Li, Youping Dong, Liping Sha, Lanjie He, Hao Wang. Inulin and Lycium barbarum polysaccharides ameliorate diabetes by enhancing gut barrier via modulating gut microbiota and activating gut mucosal TLR2+ intraepithelial γδ T cells in rats. Journal of Functional Foods, 2021, 79: 104407. DOI: 10.1016/j.jff.2021.104407.

5) The authors have to improve the legend of figure 5 to verify its conformity with actual parts of the figure and make it more clear to the reader. 

Reply: We really appreciate for your valuable comments. According to your helpful suggestion, the legend of Fig 5 has been carefully revised with the red color to ensure it more clear to the reader.

---

## [Decision Letter · Decision Letter 1]

18 Jun 2021

PONE-D-21-07383R1

Metformin intervention ameliorates AS in ApoE-/- mice through restoring gut dysbiosis and anti-inflammation

PLOS ONE

Dear Dr. Jia,

Thank you for submitting your manuscript to PLOS ONE. After careful consideration, we feel that it has merit but does not fully meet PLOS ONE’s publication criteria as it currently stands. Therefore, we invite you to submit a revised version of the manuscript that addresses the points still raised by reviewer 1.

We look forward to receiving your revised manuscript.

Kind regards,

Michael Bader

Academic Editor

PLOS ONE

Journal Requirements:

Reviewers' comments:

Reviewer's Responses to Questions

**Comments to the Author**

1. If the authors have adequately addressed your comments raised in a previous round of review and you feel that this manuscript is now acceptable for publication, you may indicate that here to bypass the “Comments to the Author” section, enter your conflict of interest statement in the “Confidential to Editor” section, and submit your "Accept" recommendation.

Reviewer #1: All comments have been addressed

Reviewer #2: All comments have been addressed

2. Is the manuscript technically sound, and do the data support the conclusions?

Reviewer #1: Yes

Reviewer #2: Yes

3. Has the statistical analysis been performed appropriately and rigorously? 

Reviewer #1: Yes

Reviewer #2: Yes

4. Have the authors made all data underlying the findings in their manuscript fully available?

Reviewer #1: Yes

Reviewer #2: Yes

5. Is the manuscript presented in an intelligible fashion and written in standard English?

Reviewer #1: Yes

Reviewer #2: Yes

6. Review Comments to the Author

Reviewer #1: Most of the raised questions have been addressed. Two minor issues left.

1. The ZO-1 quantification from immunofluorescence figures of Fig 7A looks not very consistent with the barplot of Fig 7B.

2. The subtitle of "MET attenuated the integrity of gut mucosa" should be revised as "MET restored the integrity of gut mucosa" to avoid misunderstanding.

Reviewer #2: The manuscript has been improved significantly by the authors and of the comments are properly answered.

7. PLOS authors have the option to publish the peer review history of their article (what does this mean?). If published, this will include your full peer review and any attached files.

Reviewer #1: No

Reviewer #2: No

---

## [Author Response · Author response to Decision Letter 1]

22 Jun 2021

Dear editor,

Thank you for your kind letter. Thanks for your constructive suggestions and the reviewers’ valuable comments on our manuscript. We have revised the manuscript in accordance with the comments, and carefully proofread the manuscript to minimize typographical, grammatical, and bibliographical errors.

We have uploaded 3 files (Response to Reviewers, Revised Manuscript with Track Changes, Manuscript) to the submission system. All alterations in the manuscript were marked with red color. 

Response to editor comments 

Thank you for your constructive suggestions. According to all of your kind comments, our manuscript has been carefully revised. Please check them. 

a) Journal Requirements: Please review your reference list to ensure that it is complete and correct. If you have cited papers that have been retracted, please include the rationale for doing so in the manuscript text, or remove these references and replace them with relevant current references. Any changes to the reference list should be mentioned in the rebuttal letter that accompanies your revised manuscript. If you need to cite a retracted article, indicate the article’s retracted status in the References list and also include a citation and full reference for the retraction notice.

Reply: We really appreciate your careful advice. We have carefully checked the Reference list to make sure that there’s no retracted paper in the Reference. Furthermore, we also downloaded the Plos endnote style to unify the reference format and checked the reference format manually to make sure our references style meet PLOS ONE’s publication standards. All alterations in Reference section of the manuscript were marked with red color. 

b) While revising your submission, please upload your figure files to the Preflight Analysis and Conversion Engine (PACE) digital diagnostic tool, https://pacev2.apexcovantage.com/. PACE helps ensure that figures meet PLOS requirements. To use PACE, you must first register as a user. Registration is free. Then, log in and navigate to the UPLOAD tab, where you will find detailed instructions on how to use the tool. If you encounter any issues or have any questions when using PACE, please email PLOS at figures@plos.org. Please note that Supporting Information files do not need this step.

Reply: Thank you for your comments. According to your suggestion, we have uploaded all the figures to the PACE tools, and make sure all the figures meet PLOS requirements.

Response to Reviewer #1

1) The ZO-1 quantification from immunofluorescence figures of Fig 7A looks not very consistent with the barplot of Fig 7B.

Reply: We really appreciate your careful comments. Zonula occludens-1 (ZO-1) protein, an epithelial tight junction protein, is a biomarker of intestinal permeability. With the accelerating progression of atherosclerosis, the expression of ZO-1 showed a notable decrease. In our study, in Fig 7A and 7B, compared with the CON group, the ZO-1 expression in the AS group was significantly decreased, and it could be restored by MET intervention. Thank you for your valuable advice, according to your suggestion, we have carefully checked the data in Figure 7 and found that there’s no wrong in ZO-1+ area/intestinal area in the Fig 7. The dataset for Fig 7B has been provided in the following table. Thank you for your rigorous attitudes.

ZO-1+ area / intestinal area CON AS AS+MET

 0.153 0.103 0.149

 0.149 0.114 0.158

 0.186 0.097 0.133

Mean 0.1627 0.1047 0.1467

Std. Deviation 0.02031 0.008622 0.01266

2) The subtitle of "MET attenuated the integrity of gut mucosa" should be revised as "MET restored the integrity of gut mucosa" to avoid misunderstanding.

Reply: We really appreciate for your kind comments. In our manuscript, we have replaced the subtitle of "MET attenuated the integrity of gut mucosa" as "MET restored the integrity of gut mucosa", in accordance with your constructive suggestion. The changes in the manuscript were marked with red color.

---

## [Editor Report · Decision Letter 2]

24 Jun 2021

Metformin intervention ameliorates AS in ApoE-/- mice through restoring gut dysbiosis and anti-inflammation

PONE-D-21-07383R2

Dear Dr. Jia,

We’re pleased to inform you that your manuscript has been judged scientifically suitable for publication and will be formally accepted for publication once it meets all outstanding technical requirements.

Kind regards,

Michael Bader

Academic Editor

PLOS ONE
---

## [Editor Report · Acceptance letter]

28 Jun 2021

PONE-D-21-07383R2 

Metformin intervention ameliorates AS in *ApoE**^-/-^* mice through restoring gut dysbiosis and anti-inflammation 

Dear Dr. Jia:

I'm pleased to inform you that your manuscript has been deemed suitable for publication in PLOS ONE. Congratulations! Your manuscript is now with our production department. 

Kind regards, 

on behalf of

Prof. Michael Bader 

Academic Editor

PLOS ONE